# Tri-Phenyl-Phosphonium-Based Nano Vesicles: A New In Vitro Nanomolar-Active Weapon to Eradicate PLX-Resistant Melanoma Cells

**DOI:** 10.3390/ijms26073227

**Published:** 2025-03-30

**Authors:** Silvana Alfei, Carola Torazza, Francesca Bacchetti, Maria Grazia Signorello, Mario Passalacqua, Cinzia Domenicotti, Barbara Marengo

**Affiliations:** 1Department of Pharmacy, University of Genoa, Viale Cembrano, 16148 Genoa, Italy; carola.torazza@unige.it (C.T.); francesca.bacchetti@edu.unige.it (F.B.); 2Biochemistry Laboratory, Department of Pharmacy, University of Genoa, Viale Benedetto XV 3, 16132 Genova, Italy; mariagrazia.signorello@unige.it; 3Biochemistry Section, Department of Experimental Medicine (DIMES), University of Genova, Via Alberti L.B., 16132 Genoa, Italy; mario.passalacqua@unige.it; 4Centro 3R, Department of Information Engineering, University of Pisa, Largo Lucio Lazzarino 1, 56122 Pisa, Italy; cinzia.domenicotti@unige.it; 5IRCCS Ospedale Policlinico San Martino, 16132 Genova, Italy; 6Department of Experimental Medicine (DIMES), University of Genova, Via Alberti L.B., 16132 Genoa, Italy

**Keywords:** cutaneous metastatic melanoma (CMM), triphenyl phosphonium (TPP) groups, nanosized bola-amphiphiles vesicles, cell viability (%) studies, low hemolytic effects, low cytotoxicity on HaCaT cells

## Abstract

Cutaneous metastatic melanoma (CMM) is the most aggressive form of skin cancer, with characteristics including a poor prognosis, chemotherapy-induced secondary tumorigenesis, and the emergence of drug resistance. Our recent study demonstrated that triphenyl phosphonium (TPP)-based nanovesicles (BPPB), which have amphiphilic properties, exert potent ROS-dependent anticancer effect against PLX4032 (PLX)-sensitive MeOV BRAF^V600E^ and MeTRAV BRAF^V600D^ mutant cell lines, evidencing more marked efficacy on MeOV cells. Here, taking advantage of this in vitro model, the antitumoral effect of BPPB was tested on PLX-resistant (PLX-R) MeOV BRAF^V600E^ and MeTRAV BRAF^V600D^ mutant cell lines to find a new potential strategy to fight melanoma therapy resistance. Specifically, we investigated both its effects on cell viability in dose- and time-dependent experiments and those on ROS generation. Our results show that BPPB exerted strong antiproliferative effects, regardless of their acquired resistance of cells to PLX, that correlated with ROS overproduction for 24 h treatments only. Moreover, in terms of cell viability, PLX-R MeTRAV cells demonstrated a remarkably higher tolerance to 24 h BPPB treatment than PLX-R MeOV. On the contrary, BPPB exposure for longer periods induced similar responses in both cell lines (IC_50_ = 87.8–106.5 nM on MeOV and 81.0–140.6 nM on MeTRAV). Notably, BPPB cytotoxicity on non-tumorigenic human keratinocytes (HaCaT) was low, thus establishing that BPPB is appreciably selective for CMM cells, allowing for selectivity index values (SIs) up to 11.58. Furthermore, the BPPB concentration causing 50% hemolysis (HC_50_) was found to be 16–173 and 4–192-fold higher than the IC_50_ calculated for PLX-R MeOV and MeTRAV cells, respectively. Correlation studies established that BPPB exerts cytotoxic effects on PLX-R MeOV and MeTRAV cells by a time-dependent mechanism, while a concentration-dependent mechanism was observed only at 24 h of exposure. Finally, a ROS-dependent mechanism can be assumed only in PLX-R MeTRAV cells in 72 h treatment.

## 1. Introduction

Melanoma is a malignant tumour originating from melanocytes, which are cells responsible for the synthesis of melanin, a dark pigment that is partially responsible for skin colour [1]. Melanocytes are found mostly in the skin [2], although they are also present in the eyes, ears, leptomeninges, and gastrointestinal tract, as well as in the oral, genital, and sinonasal mucous membranes. Therefore, melanoma can arise in all areas of the body where melanocytes are normally present, with a particular predilection for photo-exposed areas [2]. Although melanoma usually starts as a skin lesion, it can also grow on mucous membranes (mucosal melanoma), such as lips or genitals, and occasionally it occurs in other parts of the body such as the eyes and brain [3]. Cutaneous melanoma (CM) accounts for more than 90% of melanoma diagnoses, mainly in young-age white populations. It is particularly common among Caucasians, especially in northwestern Europeans who live in sunny places, such as Oceania, North America, Europe, South Africa, and Latin America [4]. In Italy, CM is currently the third most frequent tumour in both sexes under the age of 50. Over the last 20 years, its incidence has increased dramatically, from 6000 cases in 2004 to 11,000 in 2014. In the latest report “Cancer numbers in Italy 2024”, presented by the Italian Association of Medical Oncology, there are forecasts indicating that melanoma diagnoses, in 2024, could reach as many as 17,000, approximately 4300 more than the number recorded in 2023 (12,700). This higher number can be interpreted both as a result of a greater proclivity of the population to undergo regular checks, which are essential for an early diagnosis of the tumour, of greater exposure to risk factors, for example to solar rays without adequate protection or the use of tanning beds. Collectively, the incidence of melanoma consists of about 25 new cases per 100,000 population in Europe, 30 cases per 100,000 population in the USA, and 60 cases per 100,000 population in Australia and New Zealand [5].

### 1.1. Main Causes of CM

CM derives from the step by step gathering of genetic mutations that modify cell proliferation, differentiation, and death [6]. In white populations, CMs are estimated to be caused by the mutagenic effect of ultraviolet radiation (UVR) in more than 75% of cases [7,8]. Cutaneous metastatic melanoma (CMM) originates from a complex interaction of UVR-mediated oncogenic aberrations such as BRAF, NRAS, or KIT mutations, inherited germline genetic modifiers, such as CDKN2A, MC1R, or BAP1, and phenotypic risk factors, including lighter skin tones, sun sensitivity, or naevus count and type [2,9,10]. Moreover, immune distress due to solid organ or hematopoietic cell transplantation, other immunodeficiencies [11], and some genodermatoses, such as xeroderma pigmentosum [12], could drive to an increased risk of melanoma.

### 1.2. Possible Treatments for Melanoma

The management of melanoma mainly depends on its stage. The treatment of choice for primary melanoma implies its removal by excisional biopsy, whose margins are determined by tumor thickness. The chances of a tumour recurring or spreading depends on how deeply it has penetrated the layers of the skin. For CMM, the treatments include surgery, chemotherapy, immune checkpoint modulator therapy, and/or radiation therapy. Systemic drug therapies are advised as adjuvants to surgery in patients with resectable locoregional metastases and are crucial to the strength of treatment in advanced melanoma. Management of advanced melanoma is difficult, particularly in the case of cerebral metastasis formation. In white populations, the mortality percentage decreased by 18% within 3 years due to the development of more efficient systemic therapies, which also encompass effective treatments for asymptomatic brain metastasis [13]. Further improvement of current treatments for melanoma depends on scientists’ ability to personalize care, and multidisciplinary care is essential. Even with the rapid evolution of therapies, our best weapon against melanoma remains prevention.

### 1.3. The Study

The acquisition of drug resistance, mainly due to the constitutive activation of oncogenes, hampers the effectiveness of therapy in terms of overall survival [14,15]. Mutations in the BRAF human proto-oncogene have been identified in 50% of malignant melanomas [16], and approximately 40–70% of cases show a missense mutation, with a substitution of valine with glutamic acid at codon 600, denoted as V^600E^ [17]. The identification of oncogenic driver mutations, such as KRAS and BRAF, has driven the development of small-molecule inhibitors along the RAS-RAF-MEK-MAPK signalling pathway [18]. Currently, PLX4032 (commercially known as Vemurafenib) is approved for the treatment of patients with BRAF^V600E^-mutated CMMs [19,20] and, although less effective, of those with BRAF^V600K^, BRAF^V600R^, and BRAF^V600D^ mutations [21].

Despite the encouraging results obtained with PLX4032 (PLX), most BRAF-mutant CMM patients become drug-resistant after 6/7 months of therapy, promoting cancer relapse [22].

On this scenario, we recently reported that triphenyl phosphonium (TPP)-based nanovesicles (BPPB), having amphiphilic properties, possess ROS-dependent anticancer effects against two different PLX-sensitive BRAF mutant cell lines (MeOV BRAF^V600E^ and MeTRAV BRAF^V600D^), evidencing a more marked sensitiveness of MeOV cells.

Here, the efficacy of BPPB was tested on PLX-resistant counterparts (PLX-R MeOV and MeTRAV cells), by carrying out concentration and time-dependent cytotoxicity experiments. Moreover, since we have previously reported that the cytotoxic action of BPPB on MeOV and MeTRAV cells is accompanied by ROS overproduction, this parameter was also evaluated in BPPB-treated PLX-resistant cells. In addition, to support a possible clinical development of BPPB as an adjuvant therapy for CMM management, the cytotoxic effect of BPPB was tested on human keratinocytes (HaCaT) and on red blood cells (RBCs), and its selectivity index (SI) values were calculated. Using proper dispersion graphs and fitting them with different linear and non-linear regression models, the possible dependance of BPPB cytotoxic effects and ROS induction on BPPB concentrations and time of exposure were investigated on both CMM and HaCaT cells conducting an extensive correlation study. Finally, the possible dependance of BPPB cytotoxic effects on ROS overproduction were similarly studied.

## 2. Results and Discussion

### 2.1. 1,1-(1,12-Dodecanediyl)bis [1,1,1]-triphenylphosphonium di-bromide (BPPB)

An alkyl triphenilphosphonium derivative displaying two triphenyl cationic moieties linked to each other by a C12 alkyl chain (BPPB) was synthetized according to Figure 1, performing the procedure as recently described [23].

Associated with the numbered structure of BPPB, the list of peaks from ATR-FTIR, ^1^H, ^13^C, and ^31^P NMR spectra, as well as results from FIA-MS-(ESI) and elemental analyses of BPPB, have been included in Appendix A of our recent work [23].

### 2.2. Biological Effects of BPPB on Tumoral and Not Tumoral Human Cells Models

#### 2.2.1. The Rationale of the Study

Molecules possessing the triphenyl phosphonium (TPP) group are reported to interact electrostatically and with appreciable selectivity with the negatively-charged constituents of the cytoplasmic membrane of tumor cells. Considering this, recently, we have assayed bola-amphiphilic vesicles with two TPP groups (BPPB) on sensitive and multidrug resistant neuroblastoma (NB) cells. Subsequently, the same BPPB was tested on PLX-sensitive MeOV (BRAF^V600E^) and MeTRAV (BRAF^V600D^) metastatic melanoma cell lines, isolated from the biopsies of untreated patients [24,25]. Remarkable cytotoxic effects, which were strongly correlated with ROS production, at least in cutaneous metastatic melanoma (CMM) cells, were observed, along with low levels of toxicity vs. different mammalian cell lines and low hemolysis in red blood cells (RBC) from four healthy donors. Here, we have confirmed for the first time the cytotoxic potency of BPPB against BRAF-mutated PLX-resistant CMM [26]. Prospecting a possible clinical use on skin to treat melanoma lesions topically, BPPB selectivity for tumor cells was then assessed by evaluating its possible cytotoxicity and capability to induce ROS increase on human keratinocytes, while its hemolytic damage to RBCs from up to eight healthy volunteers was investigated. The possible existence of time-dependent, ROS-dependent, and concentration-dependent mechanisms was explored, as well.

#### 2.2.2. Concentration- and Time-Dependent Effects of BPPB on PLX-R MeOV and MeTRAV Cell Viability

Before our recent studies, which demonstrated the potent and quite selective anticancer effects of a *bis*-triphenyl phosphonium (BTPP)-bola-amphiphilic (BA) molecule (BPPB) [24,25], BA materials were extensively studied for several other applications, but not as anticancer agents *per se* [27]. Here, the effects of BPPB on cell viability were investigated on two PLX4032 (PLX)-resistant CMM cell lines (PLX-R MeOV and MeTRAV). As reported in Figure 1, BPPB caused a significant reduction in the viability of PLX-R MeOV cells starting from the lowest concentration used (0.1 µM) after 48 and 72 h of exposure, and after a shorter treatment period of 24 h when starting from 0.5 µM concentration (Figure 1a). On the contrary, BPPB caused a significant reduction in the viability of PLX-R MeTRAV cells at 0.1 µM concentration after all treatment lengths, evidencing a higher sensitivity of PLX-R MeTRAV cells as compared to PLX-R MeOV ones at the lowest concentrations (Figure 1b). Specifically, when exposed to 0.1 µM treatment, PLX-R MeTRAV cells were more sensitive than PLX-R MeOV to BPPB by 8.9, 18, and 47.8% after 24, 48, and 72 h of exposure, respectively, as demonstrated by the high viability of PLX-R MeOV cells (98.9% at 24 h, 61.1% at 48 h, and 83.6% at 72 h). Although a time-dependent cytotoxic effect was observed in both cell populations, their viability was similar after 48 and 72 h of exposure, especially when exposed to BPPB concentrations ≥0.25 and 1 µM for PLX-R MeOV and MeTRAV cells, respectively (Figure 1a,b). Collectively, PLX-R MeOV cells were more susceptible to the action of BPPB as compared to PLX-R MeTRAV ones at concentrations ≥0.5 µM (24 h) and ≥0.25 µM (48 and 72 h). In fact, after the 24 h treatment, viability in the range 33.5–70.1% was detected in PLX-R MeOV cells treated with concentrations of BPPB in the range 0.5–2 µM, while viability in the range 56.5–73.5% was observed in PLX-R MeTRAV cells exposed to the same concentrations of BPPB. Concerning treatments of 48 and 72 h in length, cell viability in the ranges 3.1–11.5% after 48 h and 4.8–7.4% after 72 h was observed for PLX-R MeOV cells, while viability ranges of 9.7–50.1% after 48 h and 9.8–32.8% after 72 h were detected for PLX-R MeTRAV cells.

In summary, PLX-R MeOV cell viability decreased to below 50% under 1.0 µM BPPB after 24 h of treatment (46.9%), and under 0.25 µM BPPB after 48 (11.5%) and 72 h (7.4%) of treatment. On the contrary, PLX-R MeTRAV cell viability remained over 50% at all BPPB concentrations tested after 24 h of exposure, while it decreased to below 50% at 0.5 µM after 48 h (26.2%) and at 0.1 µM after 72 h (43.4%). These results showed that BPPB concentrations ≥0.25 µM were able to completely kill PLX-resistant MeOV cells after 48 and 72 h treatments (4.8–11.5% viability), while BPPB concentrations ≥1 µM markedly reduced the cell viability (9.8–15.8%) of PLX-resistant MeTRAV cells after the same length of exposure. Based on this data, BPPB was remarkably more potent than self-assembled metal-phenolic nanocomplexes synthetized by Li et al., whose administration left more than 50% of B16-F10 melanoma cells derived from C57BL/6J mouse alive under up to 25 mM concentrations [28]. Using the data of cell viability as a function of BPPB concentrations, we calculated the IC_50_ of BPPB against both cell lines using GraphPad Prism software (version 8.0.1). Particularly, we first converted the bar graphs of Figure 1a,b into dispersion graphs. Then, upon conversion of µM concentrations (x) in Log_10_ (x), and using a non-linear model which considered Log _10_ BPPB concentrations vs. the normalized response (Figure 2), we derived the IC_50_ values of BPPB for both cell populations at 24, 48, and 72 h of treatment. These are reported in Table 1.

Furthermore, 72 h treatment with BPPB exerted a greater cytotoxic effect on PLX-resistant cells than the effect induced by PLX on sensitive ones. Specifically, the effect of BPPB was more cytotoxic (5-fold) on PLX-R MeOV cells than that of PLX on PLX-sensitive MeOV cells, while BPPP cytotoxicity on PLX-R MeTRAV cells was 231-fold higher than that of PLX on PLX-sensitive MeTRAV cells [26].

##### Correlation Between BPPB Cytotoxic Effects and BPPB Concentrations

Here, using the dispersion graphs of cell viability (%) of both PLX-R MeOV and MeTRAV cells after 24, 48, and 72 h of exposure to BPPB vs. BPPB concentrations administered and their best statistical models, we established if it is possible to hypothesize a concentration-dependent mechanism for the cytotoxic effects of BPPB. To this end, we investigated the existence or absence of a significant correlation between BPPB anticancer effects and the administered concentrations, and which type of correlation it could be, on the basis of the R-square (R^2^) values of the statistical models which best fitted the data of dispersion graphs. R^2^ is an index that measures the goodness of fit of a statistical model applied to two series of data, and its capability to correctly predict a new response variable “y” (cell viability %) from experimental data “x” (BPPB concentration) [29]. The higher the value of R^2^, the higher the goodness-of-fit of the regression model used, and the higher the correlation existing between the two series of data fitted with that model. Low values of R^2^ for the best-fitting regression models indicate a low reliability of the models and a low correlation between data. In scientific studies, the R^2^ of a model should be >0.95 to consider that model reliable and to assert the existence of a certain correlation between data [30]. For nonlinear models, R^2^ can be misleading and may not accurately reflect the performance of the model. This is because R^2^ is defined for linear models, and if applied to nonlinear ones, it may produce values outside the range 0–1, making it difficult to interpret [31]. In our case, the R^2^ values provided by Microsoft Excel 365 software for the nonlinear models used were all in the range 0–1, so we considered R^2^ as a reliable parameter for judging the goodness of fit of the tested regression models. Appendix A shows the dispersion graphs of cell viability of PLX-R MeOV (blue square (24 h), pink triangular (48 h), and green round (72 h) indicators) and of PLX-R MeTRAV (red square (24 h), purple triangular (48 h), and sky blue (72 h) indicators) cells vs. BPPB concentrations (0.1–2.0 µM). Additionally, it reports the nonlinear regression models (dotted lines), which, among others tested, best fitted the data of dispersion graphs according to R^2^. Appendix A shows also the equations and values of R^2^ related to the selected models. The nonlinear regression models were constructed using Microsoft Excel 365 software. Values of R^2^ > 0.95 (0.9869 (MeOV) and 0.9633 (MeTRAV)) were obtained only for the statistical models used to describe the data of cell viability vs. BPPB concentrations of both cell lines after 24 h treatments, thus establishing the existence of a good correlation between data. In particular, for PLX-R MeOV cells, a polynomial correlation of second order exists, and for PLX-R MeTRAV cells, a logarithmic correlation exists. R^2^ values slightly less than 0.95 (0.9370 (48 h) and 0.9228 (72 h)) were obtained for the statistical models used to describe the data for PLX-R MeTRAV cells over longer treatment periods, establishing the existence of a poor correlation between them. R^2^ values of 0.8491 (48 h) and 0.8441 (72 h) were obtained for the statistical models used to describe the data for PLX-R MeOV cells over these time periods, establishing the absence of correlation between them. Collectively, this investigation proves that the anticancer effects of BPPB on both cell lines are dependent on its concentrations only for treatments of 24 h in length. When cells were treated for longer periods, the dependence of BPPB’s cytotoxic effects on its concentration was scarce for PLX-R MeTRAV cells, and absent for PLX-R MeOV ones.

##### Correlation Between BPPB Cytotoxic Effects and Time of Exposure

Here, following the same method used in the previous Section, we assessed the existence or absence of a certain dependence of BPPB anticancer effects on times of treatments. As shown in Appendix A, on the basis of the very high R^2^ values (both equal to 1) of the nonlinear regression models that best fitted the dispersion graphs of the IC_50_ values (µM) vs. times of exposure (hours) obtained for PLX-R MeOV cells (blue square indicators) and PLX-R MeTRAV ones (pink triangular indictors), it was found that a strong correlation of second order polynomial type exists between the data. Collectively, although not linear, the existence of a correlation between the cytotoxic effects of BPPB and the duration of treatments established the existence of a certain time-dependent mechanism in the action of BPPB. Despite the existence of a strong correlation between the IC_50_ values of BPPB and the exposure timing, these findings were nonetheless different from those observed when BPPB was tested on CMM cells sensitive to PLX [24]. Indeed, in that case, a linear correlation was accepted on the basis of R^2^ values of the linear regression models obtained using similar graphs (0.9754 (MeOV) and 0.9190 (MeTRAV)). Furthermore, the dispersion graphs in Appendix A clearly show that in experiments of 24 h in length, PLX-R MeTRAV cells were 3.8 times more tolerant to BPPB than PLX-R MeOV ones, while at both 48 and 72 h of exposure the IC_50_ values of BPPB were similar, and no significant difference was observed in the responses of PLX-R MeOV and MeTRAV, regardless of their different BRAF mutations.

#### 2.2.3. Concentration- and Time-Dependent Effects of BPPB on ROS Production in PLX-R Cells

TPP-based compounds, including TPP-BA as BPPB, have been found to induce an overproduction of ROS as a marker of oxidative stress (OS), which can be responsible for the cytotoxic effects of such compounds [24,32]. In this regard, we recently reported that BPPB induced a decrease in MeOV and MeTRAV cell viability that was linearly correlated with an increase in ROS production [24]. Therefore, ROS levels were analogously analyzed also in PLX-R MeOV and MeTRAV cell populations treated with BPPB, to confirm or confute the previous findings.

As shown in Figure 3a,b, after 48 and 72 h, BPPB induced a significant concentration- and time-dependent increase in ROS production in both cell populations, while no significant changes were observed in cells treated for 24 h. Collectively, ROS production was slightly higher in PLX-R MeOV than in PLX-R MeTRAV PLX-R cells, showing an opposite trend to that observed for cell viability. In fact, when PLX-R cells were exposed to BPPB for only 24 h, low ROS production in both cell populations was accompanied by high cell viability, especially in PLX-R MeTRAV cells. Conversely, over longer times of exposure, cell viability became remarkably lower, and ROS production was significantly higher with respect to the control. Based on these findings, a possible ROS-dependent mechanism for BPPB effects could be hypothesized.

##### Correlation Between ROS Production Increase and BPPB Concentrations

Here, we investigated the existence of a possible correlation between the increasing production of ROS and BPPB concentrations, and thus our capability to assume a concentration-dependent mechanism. Appendix A reports the dispersion graphs of the DCFH positive cells (%) vs. BPPB concentrations (µM) (round indicators without lines) for both cell lines after 24, 48, and 72 h of treatments. The related best-fitting nonlinear regression models, their equations, and the connected R^2^ values were provided via Microsoft Excel 365, to assess the possible existence of a certain type of correlation between the two series of data. Values of R^2^ > 0.95 [30] would have confirmed the existence of a correlation and of a concentration-dependent mechanism. Appendix A reveals that, regardless of their values of R^2^, the nonlinear regression models which best described the data of dispersion graphs were of second order polynomial type for data for experiments on PLX-R MeOV cells at 24 h of exposure and on PLX-R MeTRAV cells at 72 h of exposure. Logarithmic nonlinear statistical regressions were the models which best fitted experimental data obtained for PLX-R MeOV cells at 48 and 72 h of exposure, while statistical models of Power type best fitted the experimental data obtained for PLX-R MeTRAV cells at 24 and 48 h of exposure. Only the R^2^ value of the nonlinear regression model which best fitted the data of the dispersion graph obtained in experiments on PLX-R MeTRAV cells when treated with BPPB for 72 h was >0.95 (0.9799), thus confirming the existence of a positive correlation of second order polynomial type between increasing ROS production and BPPB concentrations (µM). In all other cases, the R^2^ values of the best-fitting nonlinear regression models were <0.95, thus establishing the absence of a significant correlation of any type between increasing ROS production and BPPB concentrations (µM). Collectively, only for BPPB ROS induction in PLX-R MeTRAV cells treated for 72 h is it possible to claim a concentration-dependent mechanism. The dependence of BPPB ROS induction in PLX-R MeTRAV cells treated for 48 and 24 h ranged from scarce to very poor (R^2^ = 0.9228 and 0.9011, respectively), and this dependence was also scarce in PLX-R MeOV cells when treated for 24 h with BPBB (R^2^ = 0.9324). Finally, no ROS dependence on BPPB concentrations exists for BPPB ROS induction in PLX-R MeOV cells when treated for 48 h.

##### Correlation Between ROS Production Increase and Exposure Timing

Appendix A reports the dispersion graphs of the average ROS production, expressed as DCFH positive cells (%) in both PLX-R MeOV and MeTRAV cells vs. times of exposure tested. The related logarithmic regression models which best fitted the dispersion graphs and the connected R^2^ values were also provided via Microsoft Excel 365 to assess the possible existence of a type of correlation between the two series of data. Appendix A demonstrates that the average ROS production in PLX-R MeOV cells was higher than that in PLX-R MeTRAV ones by 1.3 (24 h), 2.0 (48 h), and 2.3 times (72 h). A very high logarithmic correlation was found between the average ROS production and exposure timing in both types of cells, thus confirming that ROS production was dependent on the time of treatments. Moreover, the average DCFH positive cells (%) increased by 5.9 and 8.6 times, respectively, from 24 to 48 h and then to 72 h exposure, with the increase from 48 to 72 h being 1.5 times when PLX-R MeOV CMM cells were treated. Conversely, it increased only by 3.7 and 4.7-times, respectively, from 24 to 48 h and then to 72 h of exposure, with an increase of 1.3 times from 48 to 72 h when PLX-R MeTRAV cells were exposed to BPPB.

##### Correlation Between BPPB Cytotoxic Effects (Cell Viability %) and ROS Overproduction

As evidenced in Section 2.2.3, the trend observed for ROS overproduction was like and opposite that observed for cell viability (%) as obtained in the cytotoxicity experiments, thus envisaging a possible correlation between the cytotoxicity effects of BPPB and its capability to induce ROS production, and therefore a ROS-induced anticancer mechanism. To confirm this assumption, we carried out experiments like those performed in the previous Sections for other series of data. Appendix A reports the dispersion graphs of the DCFH positive cells (%) vs. cell viability (%) (round indicators without lines) observed at the same concentrations of BPPB for both cell lines after 24, 48, and 72 h of treatment. The related best-fitting nonlinear regression models, their equations, and the connected R^2^ values were also provided via Microsoft Excel 365 to assess the possible existence of a certain type of correlation between the two series of data. Values of R^2^ > 0.95 would have confirmed the assumed correlation. Appendix A highlights that, regardless of their values of R^2^, the nonlinear regression models which best described the data of the dispersion graphs were of Power type for experiments on both cell lines at 24 and 72 h of exposure, while they were of second order polynomial type when the duration of treatments was 48 h, thus establishing a similar trend for both PLX-R MeOV and MeTRAV cells. Only the R^2^ value of the nonlinear regression model which best fitted the data of the dispersion graph obtained in experiments on PLX-R MeTRAV cells treated with BPPB for 72 h was >0.95 (0.9694), thus confirming the existence of an inverse correlation of Power type between ROS hyperproduction and cell viability (%). In all other cases, R^2^ values of the best-fitting nonlinear regression models were <0.95, thus establishing the absence of a significant correlation of any type between ROS hyperproduction and cell viability (%). Collectively, only for the cytotoxic effects of BPPB on PLX-R MeTRAV cells treated for 72 h was it possible to claim a ROS-dependent mechanism. The dependence of the cytotoxic effects of BPPB on PLX-R MeTRAV cells treated for 24 and 48 h ranged from scarce to poor (R^2^ = 0.9305 and 0.9193, respectively), while no ROS dependence existed for BPPB cytotoxic effects on PLX-R MeOV cells. From these early experiments, possible oxidative cell death can be hypothesized for PLX-R MeTRAV cells after 72 h exposure. Despite MeOV and MeTRAV cells being sensitive to PLX, a ROS-dependent cytotoxic effect was established for both cell lines. It was found to be stronger for MeTRAV cells as compared to MeOV ones [24].

#### 2.2.4. In Vitro Hemolytic Toxicity of BPPB on Red Blood Cells (RBC)

The hemolytic ratio percentage (%) caused by BPPB was assessed as recently reported with slight changes [33]. Particularly, EDTA-blood samples from eight healthy donors were exposed to increasing concentrations of BPPB (0.1–50 µM). The results have been expressed as an average of eight independent determinations ± S.D. and are shown in the form of a bar graph of these means vs. BPPB concentrations in Figure 4a, and in the form of a dispersion graph (pink square indicators) in Appendix A. In Appendix A, the best-fitting nonlinear statistical model related to the dispersion graph (punctuated pink line) with its equation and R^2^ value, as obtained using Microsoft Excel 365 software, was also included. Figure 4b reports the trend of the viability (%) of RBC, PLX-R MeOV, and PLX-R MeTRAV cells vs. BPPB concentrations in the range 0.1–2.0 µM, as observed in cytotoxicity experiments.

As observable, hemolysis was statistically significant with respect to control (CTRL) only for concentrations ≥10 µM, but was limited to 23.8% for BPPB = 10 µM. A concentration of BPPB = 20 µM was necessary to determine a hemolysis rate of slightly higher than 50% (50.3%). Based on the R^2^ value being greater than 0.95 (0.9617), we can assert that the decrease in RBC viability (%) and consequently the increase in hemolysis is correlated to the increasing BPPB concentration (Appendix A). Specifically, a second order polynomial correlation was detected between RBC viability and BPPB concentration, thus establishing a concentration-dependent mechanism for the observed hemolysis. Figure 4b shows that at the highest concentration tested on both CMM cells (2.0 µM), RBC viability (%) was still 98.03%, while that of PLX-R MeOV cells was 33.48% (24 h), 4.97% (48 h), and 4.83% (72 h), and that of PLX-R MeTRAV cells was 56.51% (24 h), 9.67% (48 h), and 9.78% (72 h). As previously described for viability experiments carried out with BPPB on CMM cells, using data reported in the bar graph (Figure 4a), a dispersion graph of BPPB concentrations vs. RBC viability was created and was used to calculate the HC_50_ value of BPPB on RBCs (intended as the concentration of BPPB needed to cause 50% hemolysis) using GraphPad Prism 8.0.1 Software (GraphPad Software, Boston, MA, USA). Briefly, the data representing RBC viability (%) vs. BPPB concentrations were transformed into data representing RBC viability vs. Log_10_ BPPB concentrations (green indicators with error bars in Figure 5). The plot of nonlinear regression of Log_10_ concentrations of BPPB vs. normalized response (green trace without indicators in Figure 5) was obtained using GraphPad Prism 8.0.1 Software (GraphPad Software, Boston, MA, USA) and used to calculate the HC_50_, which was 15.56 ± 12.13 µM.

#### 2.2.5. Concentration and Time-Dependent Effects of BPPB on HaCaT Cell Viability

To ensure possible clinical developments of BPPB as a topical drug for treatment of melanoma skin lesions, it was necessary to ascertain its effects on non-cancerogenic dermal cells. Therefore, HaCaT cells, an immortal non-cancerous keratinocyte cell line originated from adult human skin [34,35,36,37], were treated with BPPB (0.1–10 µM for 24, 48, and 72 h) and then analyzed in terms of cell viability. Note that the cytotoxic effects of BPPB on other mammalian cells, including human hepatocytes (HepG2), monkey kidney cells (Cos-7), and human fibroblasts (MRC-5), were investigated in our previous study published in Nanomaterials (2024) [25], and the results are available in Appendix A. Since the outermost layer of human skin is made up of 90% keratinocytes and the remaining 10% is melanocytes (from which melanoma can originate), we considered here that the use of human keratinocytes to test the cytotoxicity of BPPB could be the most appropriate choice, when considering the potential future clinical development of a topical treatment to be applied to the skin to treat melanoma lesions. The results of cell viability (%) in the treatment of HaCaT cells with increasing BPPB concentrations at all times tested have been reported as bar graphs in Figure 6a and as dispersion graphs in Figure 6b, as well as in Appendix A. In Appendix A, the best-fitting nonlinear statistical models related to the dispersion graph (punctuated lines) with their equations and R^2^ values as obtained using Microsoft Excel 365 software are also included. Furthermore, Figure 7 reports a comparison between HaCaT, PLX-R MeOV, and MeTRAV cell viability (%) vs. BPPB concentration in the range 0.1–2.0 µM.

As shown in Figure 6a,b, 24 h of BPPB treatment significantly reduced cell viability. However, it remained >50% (59.80%) up 6.0 µM BPPB concentration, while it dramatically decreased to 10.08% at 10.0 µM, the highest concentration tested. A similar trend was observed also at 48 and 72 h. Although HaCaT cell response to BPPB was very similar over 48 and 72 h treatments, data recorded after all exposure timings highlighted that there is a threshold concentration over which BPPB was dramatically lethal (Figure 6a,b). Based on the R^2^ values of the best-fitting nonlinear models shown in Appendix A, a second order polynomial correlation between cell viability (%) and BPPB concentrations is observable only in the 24 h experiments (R^2^ > 0.95). The correlation (logarithmic type) was scarce at 72 h treatment (R^2^ = 0.9324), while no correlation existed (R^2^ = 0.8414) in the 48 h experiments, thus establishing that a concentration-dependent mechanism is conceivable only for short 24 h treatments, as observed also for tumoral cell populations considered here. Collectively, even if less tolerant to BPPB than RBCs, HaCaT cells were much more tolerant than MeOV and MeTRAV cells, regardless of their resistance to PLX (Figure 7), thus allowing the determination of a certain selectivity of BPPB for CMM cells and illustrating their promising therapeutic potential. To confirm this assumption and to allow a direct comparison of the cytotoxic effects of BPPB observed towards HaCaT cells and those observed vs. CMM cells, as well as comparisons with the cytotoxic effects of other synthetized nanoparticles reported in the literature, the IC_50_ of BPPB vs. HaCaT was calculated. As previously described for viability experiments carried out with BPPB on CMM cells and RBCs, the transformed dispersion graph of Log_10_ BPPB concentrations vs. HaCaT cell viability (%) and the plots of nonlinear regression of Log_10_ concentrations of BPPB vs. normalized response reported in Figure 8 were used to calculate the IC_50_ value of BPPB on HaCaT.

Graphs in Figure 8 and the IC_50_ results reported in Table 2 were achieved using GraphPad Prism 8.0.1 Software (GraphPad Software, Boston, MA, USA). Table 2 reports also the IC_50_ values of BPPB vs. PLX-R MeOV and MeTRAV cells, as well as HC_50_ values vs. RBCs, for direct comparisons.

The IC_50_ values reported in Table 2 confirmed that at all times of exposure tested, HaCaT cells were more tolerant to BPPB than both melanoma cell populations, and especially at 48 and 72 h. Additionally, BPPB cytotoxicity towards HaCaT cells was more minor than that of cationic dendrimer nanoparticles (PAMAM) of fourth (G4), fifth (G5), and sixth (G6) generations, as observed by Mukherjee et al. after 24 h of exposure [37]. Specifically, BPPB was less toxic than G4, G5, and G6 by 1.3, 3.8, and 3.9 times, respectively. Curiously, as observed for PLX-R MeOV cells, but to a more major extent, the IC_50_ values calculated for HaCaT were higher after 72 h of exposure than after 48 h of treatment, thus forecasting a sort of adaptation of these cells over time to the cytotoxic effects of BPPB. The trend of the cytotoxic effects of BPPB vs. HaCaT as a function of exposure timing is shown in Appendix A. Appendix A reports also the best-fitting nonlinear statistical models related to the trend dispersion graphs (punctuated lines) with their equations and R^2^ value, as obtained using Microsoft Excel 365 software. Based on the R^2^ value (R^2^ = 1), a strong second order polynomial correlation exists between the IC_50_ values of BPPB and exposure timing, thus establishing the existence of a time-dependent mechanism, as previously observed on both PLX-R cells.

#### 2.2.6. Concentration and Time-Dependent Effects of BPPB on ROS Production by HaCaT Cells

As for PLX-R MeOV and MeTRAV cells, ROS levels were analyzed in HaCaT cells exposed to increasing concentrations of BPPB (Figure 9a).

As shown in Figure 9a, the increase in ROS levels was significant and very strong only at the highest concentrations tested (10.0 µM) after 24 and 48 h of exposure (ratios = 10.25 and 12.67, respectively) and at 6.0 and 10.0 µM after 72 h (ratios = 9.13 and 8.92, respectively). At the highest concentration, the ROS amount after 72 h of exposure was lower than after 48 h, resulting in a hypothesis of cellular adaptation. Collectively, this response is similar to that observed for PLX-resistant cells, in which an inverse relationship between cell viability and ROS production has been reported (Figure 9b). Therefore, it is conceivable that the BPPB cytotoxic effects could depend on ROS production increase. This assumption found confirmation in the modelling data of HaCaT cell viability (%) vs. the DCFH positive cells (%) after exposure to increasing concentrations (0.1–10.0 µM) of BPPB for 24, 48, and 72 h, with the best-fitting regression models shown in Appendix A. Based on the high R^2^ values (0.9874 (24 h), 0.9967 (48 h), 0.9927 (72 h), a strong correlation of Power type was found to exist between cell viability (%) and ROS increase. Dispersion graphs and associated best-fitting nonlinear models (Appendix A) were elaborated to evidence the possible existence of a concentration or time-dependent mechanism for ROS increase. Based on the R^2^ values reported (0.9685 (24 h), 0.9669 (48 h), 0.9345 (72 h), a good second order polynomial correlation between ROS increase and BPPB concentrations existed only for treatments lasting 24 and 48 h, for which it is possible to confirm a concentration-dependent mechanism, while for treatments lasting 72 h, the correlation was scarce. The increase in ROS levels was strongly correlated to exposure timing, thus establishing a time-dependent mechanism for ROS production induced by BPPB.

#### 2.2.7. Selectivity Index

The selectivity index (SI) of BPPB for CMM cells vs. RBCs and HaCaT cells was calculated to predict its therapeutic potential. Generally, the SI values against tumoral cells (TCs) in relation to non-tumoral cells (NTCs) is calculated using the Formula (1).SI = IC_50_ for NTCs/IC_50_ for TCs(1)

A sufficiently high value of the SI is an essential requirement to make a new molecule worthy of consideration for further studies and future development as a new therapeutic agent. Generally, tested compounds with selectivity indices higher than 1 indicate drugs with greater efficacy against tumor cells than toxicity against normal cells [38]. Although opinions are contrasting, Krzywik et al. have recently reported that SIs > 1.0 can be considered sufficient and favorable for the development of a new molecule as chemotherapeutic [38].

The IC_50_ values reported in Table 2, determined after 24, 48, and 72 h of exposure of CMM and HaCaT cells to BPPB, as well as the HC_50_ determined on RBCs after the time of experiments according to the protocol in [33], were used to calculate the SIs according to Equation (1). These are reported in Table 3.

Concerning RBCs, the selectivity of BPPB for tumor cells ranged from good to very high for both cell lines and at all times of exposure tested. Specifically, SI values were under 20 (4.10 vs. PLX-R MeOV and 14.46 vs. PLX-R MeTRAV cells) for exposure timing of 24 h, while for longer periods of exposure they were >100, up to 177.22 (48 h, PLX-R MeOV cells) and 192.10 (72 h, PLX-R MeTRAV cells). Interestingly, the selectivity of BPPB for PLX-R MeTRAV cells compared to RBCs after 72 h of exposure was higher than that of the *Pulsatilla saponin D* derivative **14** reported by Zhong et al., which was among the most active and least toxic anticancer devices against the A549 human lung cancer cell line [39]. In fact, by using the reported data of IC_50_ on A549 cells after 72 h of exposure and HC_50_ on RBCs reported for compound **14**, a SI < 192.10 and equal to 178.57 can be calculated [39]. Regarding all other compounds, two demonstrated an SI < 10 (1.05 and 7.3), eight ≤ 50 (20, 23.9, 29, 30.8 and 50), two < 100 (80 and 98), and only one > 100 (108), which was lower by 1.4 and 1.8 times than those calculated for BPPB when assayed on PLX-R MeOV and MeTRAV cells [39]. Curiously, while the SI values vs. PLX-R MeTRAV cells were 3.5 and 1.6 times lower than PLX-R MeOV cells when exposed to BPPB for 24 and 48 h, they were higher by 1.3 times when exposed for 72 h. Practically, longer periods of exposure reduced (48 h) and even reversed (72 h) the differences in cell tolerability to BPPB, as observed also for HaCaT cells both in terms of cell viability and ROS production. This trend has been shown in Figure 10. The graph reports the SI values of BPPB for both PLX-R MeOV and MeTRAV cells in relation to its hemolytic toxicity as a function of length of exposure.

Concerning HaCaT cells, the selectivity of BPPB for tumor cells was from >1 up to 11.4 for both CMM cell lines and at all times of exposure tested. Specifically, SI values were >>>1 vs. PLX-R MeOV (4.00 (24 h), 7.95 (48 h), and 8.67 (72 h)) with a two-fold increase between 24 and 48 h of exposure, and a further 1.1 times increase at 72 h exposure. Considering PLX-R MeTRAV cells, although the SI was only 1.06 for short times of exposure (24 h), it progressively increased by 4.7 and a further 2.3 times for longer exposure timings of 48 and 72 h. Collectively, after 72 h exposure, BPPB demonstrated a selectivity for both PLX-resistant melanoma cells tested here that can be considered higher than that observed by Wróblewska-Łuczka for four out of five terpenes for four melanoma cell lines (A375, SK-MEL 28, FM55P and FM55M2), as these exhibited SI values under 4 [40]. Only Carvacrol was found to possess a selectivity index above 10, as in the case of BPPB when it was administered in PLX-R MeTRAV cells [40]. The trend of the SI values of BPPB for both PLX-R MeOV and MeTRAV cells in relation to its cytotoxicity vs. HaCaT cells as a function of length of exposure is available in Figure 10.

## 3. Materials and Methods

### 3.1. Chemicals and Instruments

All reagents and solvents used in this study were obtained from Merck (Milan, Italy) and were used without further purification. 1,1-(1,12-dodecanediyl)bis [1,1,1]-triphenylphosphonium di-bromide (BPPB) was synthetized and characterized as recently described [23].

### 3.2. BPPB Cytotoxicity Evaluation on PLX-Resistant CMM Cells

#### 3.2.1. Cell Lines and Culture Conditions

PLX4032-resistant (PLX-R) MeOV and MeTRAV cells were selected by treating the whole population of parental cells with increasing concentrations of PLX4032, as previously reported [26]. Both cell lines were maintained in RPMI 1640 medium (Euroclone Spa, Pavia, Italy) supplemented with 10% fetal bovine serum (FBS, Euroclone Spa, Pavia, Italy), 1% L-Glutamine (Euroclone Spa, Pavia, Italy), and 1% penicillin/streptomycin (Euroclone Spa, Pavia, Italy) and grown in standard conditions (37 °C humidified incubator with 5% CO_2_).

#### 3.2.2. Treatments

To determine the cytotoxic effects of BPPB to MMCs, PLX-R MeOV and MeTRAV cell lines were treated for 24, 48, and 72 h with increasing concentrations (0.1–2.0 µM) of BPPB. The stock solutions of these compounds were prepared in 40,000-fold diluted DMSO, and preliminary experiments demonstrated that the final DMSO concentrations did not change any of the cell responses analyzed. Cell cultures were carefully monitored before and during the experiments to ensure optimal cell density. Notably, samples were discarded if the cell confluence reached >90%.

#### 3.2.3. Cell Viability Assay

Cell viability was determined by using the CellTiter 96^®^ AQueous One Solution Cell Proliferation Assay (Promega, Madison, WI, USA), as previously described [41,42]. Briefly, cells (10,000 cells/well) were seeded into transparent plastic 96-well plates (Corning Incorporated, Corning, NY, USA) and then treated. Next, cells were incubated with 20 µL of CellTiter, and the absorbance at 490 nm was recorded using a microplate reader (EL-808, BIO-TEK Instruments Inc., Winooski, VT, USA). The cell survival rate, expressed as cell viability percentage (%), was evaluated based on the experimental outputs of treated groups vs. untreated groups (CTR) and was calculated as follows: cell viability (%) = (OD treated cells − OD blank)/(OD untreated cells − OD blank) × 100%.IC_50_ was evaluated by GraphPad Prism 8.0.1 Software (GraphPad Software, Boston, MA, USA) as explained in the Section 2.

### 3.3. In Vitro Hemolytic Toxicity of BPPB Using Red Blood Cells (RBCs)

The hemolytic ratio in EDTA-blood samples from eight healthy donors from the San Martino Hospital Transfusion Centre was evaluated as recently reported, with some slight changes [33]. In detail, red blood cells (RBCs) were isolated by diluting 0.2 mL of blood with 0.4 mL of phosphate-buffered saline (PBS) and centrifuged for 5 min at 10,000× *g*. The pellet, consisting of RBC, was washed five times with 1.0 mL PBS and finally resuspended with 2.0 mL PBS. The assay was carried out on 0.1 mL of resuspended RBCs added to 0.1 mL of H_2_O (positive control), 0.1 mL of PBS (negative control), or 0.1 mL of the different concentrations (0.1–50 µM) of the compound to be tested (BPPB). The samples were incubated for 60 min at 37 °C and then centrifuged for 5 min at 10,000× *g*. Finally, 0.1 mL of supernatant was transferred to a transparent 96-well plate and measured by a spectrophotometer (Dynex Technologies; Chantilly, VA, USA) at 595 nm. The hemolytic ratio was calculated using the formula:Hemolytic ratio (%) = (OD _TEST_ − OD _NEGATIVE CONTROL_)/(OD _POSITIVE CONTROL_ − OD _NEGATIVE CONTROL_) × 100

### 3.4. Evaluation of Cytotoxicity of BPPB on Human Keratinocites (HaCaT)

#### 3.4.1. Cell Culture

Human skin keratinocytes cells (HaCaT), obtained thanks to a generous gift from the IRCCS Ospedale Policlinico San Martino, Proteomics and Mass Spectrometry Unit (Genoa, Italy), were maintained in DMEM high glucose medium (Euroclone Spa, Pavia, Italy) supplemented with 10% fetal bovine serum (FBS, Euroclone Spa, Pavia, Italy), 1% L-Glutamine (Euroclone Spa, Pavia, Italy) and 1% penicillin/streptomycin (Euroclone Spa, Pavia, Italy) and grown in standard conditions (37 °C humidified incubator with 5% CO_2_).

#### 3.4.2. Treatments

To determine the cytotoxic effects of BPPB on human keratinocytes (HaCaT), cells were treated for 24, 48, and 72 h with increasing concentrations (0.1–10.0 µM) of BPPB. Cell cultures were carefully monitored before and during the experiments to ensure optimal cell density. Notably, samples were discarded if the cell confluence reached >90%.

#### 3.4.3. Viability Assay

A viability assay on HaCaT cells was performed as described in Section 3.2.3.

### 3.5. ROS Production

ROS production was evaluated using 2′–7′-dichlorofluorescein-diacetate (DCFH-DA; Merk Life Science S.r.l. Milan, Italy) as previously reported [41,42,43]. Briefly, melanoma cells (10^4^ cells/well) were seeded in 96-well plates (Corning) and treated. Then, cells were stained with 2′–7′ dichlorofluorescein-diacetate (DCFH-DA; Sigma-Aldrich, Milan, Italy) and incubated with 90% DMSO for an additional 10 min in the dark. The generated fluorescence intensity was monitored with a Perkin Elmer fluorometer (Perkin Elmer Life and Analytical Sciences, Shelton, WA, USA) at 485/530 nm excitation/emission. After the measurements, the standardization of ROS production as a function of cell number was necessary. Fluorescence values were normalized to the protein content to avoid data misinterpretation due to variations in cell numbers [44].

### 3.6. Statistical Analyses

Results have been expressed as means ± S.D. of four independent experiments in which different wells were analyzed every time for each experimental condition. In the analysis of cell viability or H_2_O_2_ levels, the condition of untreated cells was set as 100% ± S.D. and 1 ± S.D., respectively. Statistical significance of differences was determined by one-way analysis of variances (ANOVA) followed by Dunnet’s multiple comparison test correction using GraphPad Prism 8.0.1 (GraphPad Software v8.0, San Diego, CA, USA). Asterisks indicate the following *p*-value ranges: * *p* < 0.05, ** *p* < 0.01, *** *p* < 0.001, **** *p* < 0.0001. *p* > 0.05 was not considered statistically significant and no symbol was used in the images.

## 4. Conclusions

Systematic studies on genetic alterations in human malignancies have supported the development of genotype-driven targeted drugs for several types of cancers, including cutaneous melanoma. Nonetheless, the occurrence of acquired resistance to these anti-cancer agents poses a critical obstacle for improving cancer patient prognosis. In fact, while the inhibition of BRAF or of combined BRAF/MEK initially reduced the tumor progression in BRAF^V600E^ mutant melanoma patients, the targeted therapy resistance emerged in most of them within 1 or 2 years, via several mechanisms. This is the case with PLX, which improves disease and overall survival in inoperable advanced BRAF^V600E^ mutant melanoma patients, but its tolerability is often poor, and resistance frequently occurs, without any predictive factors to consider. Collectively, the current worrisome scenario concerning melanoma relates to a therapeutic arsenal that is no longer effective against resistant BRAF mutant metastatic melanoma and associated limited possibilities to improve patients’ conditions and rates of survival. Here, we demonstrated that synthetized triphenyl phosphonium cationic nano vesicles (BPPB), which have been previously found to be effective on two populations of BRAF mutant metastatic melanoma cells still sensitive to PLX (MeOV BRAF^V600E^ and MeTRAV BRAF^V600D^ cells), are effective also on the PLX-R counterpart of cells (PLX-R MeOV BRAF^V600E^ and MeTRAV BRAF^V600D^ cells) at nanomolar concentrations after 48 and 72 h of treatment. Overall, when used to treat PLX-R cells for 72 h, BPPB was more potent than PLX when used to treat PLX-sensitive cells over the same time of exposure. Collectively, although further experiments are needed, these promising early findings could pave the way for the future clinical development of BPPB as a topical drug to treat skin melanoma lesions, due to its good tolerability by dermal cells.

## Data Availability

All data supporting reported results are included in this manuscript.

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
