# Peer review of "Tri-Phenyl-Phosphonium-Based Nano Vesicles: A New In Vitro Nanomolar-Active Weapon to Eradicate PLX-Resistant Melanoma Cells"

_ijms, 2025, doi:10.3390/ijms26073227_

Round 1
Reviewer 1 Report
Comments and Suggestions for Authors
General comments: The article comes across the research and statistical analyses of the obtained results concerning the development of the new antumelanoma strategy with using the BPPB. I found this article as interesting and its topic suitable to publish in IJMS. However, I have some concerns and suggest some modification in the main body text.
Line 23; It is mentioned that: “Here, BPPB was tested on in vitro selected vemurafenib (PLX)-resistant MeOV 23 BRAFV600E and MeTRAV BRAFV600D mutants cell lines”. However there is no mentioned what exactly was testes. Please, rephrase it.
Line 26; “demonstrated a remarkable higher tolerance” of what?
Line 29; “allowing for selectivity index 29 values (SIs) up to 11.58; instead, the BPPB concentration causing 50% hemolysis (HC50) was found 16-173 and 4-192-fold higher than the IC50 calculated for MeOV and MeTRAV PLX-R cells, respectively” The sentence is not undesrstandable. What for is the Sls index here? How it correlates to HC50 value, or maybe to IC50 value? How to interpret the number results: 16-173 and 4-192. This part must be rewritten.
Line 181, 302; There is lack in the caption figure meaning of the dots on the graphs.
Line 449; There is no graph a) and c). What more, this section, regarding the cytotoxicity should have been displayed after the figure 1, where the cell viability of MeOV and MeTRAV PLX-R CMM was analyzed. It is mandatory to compare the cytotoxic efrect either to cancerogenic or normal cell line.
Line 663; It is lack of the describing in details how the ROS production assay was performed. Explain it.
Author Response
General comments: The article comes across the research and statistical analyses of the obtained results concerning the development of the new antumelanoma strategy with using the BPPB. I found this article as interesting and its topic suitable to publish in IJMS. However, I have some concerns and suggest some modification in the main body text.
We thank a lot the Reviewer for appreciating our work. We therefore will try to satisfy all her/his requests to further improve its quality.
Line 23; It is mentioned that: “Here, BPPB was tested on in vitro selected vemurafenib (PLX)-resistant MeOV 23 BRAFV600E and MeTRAV BRAFV600D mutants cell lines”. However there is no mentioned what exactly was testes. Please, rephrase it.
We thank a lot the Reviewer for her/his suggestion. The missing information has been included in lines 23-28.
Line 26; “demonstrated a remarkable higher tolerance” of what?
On suggestion of the Reviewer, this statement has been clarified. Please, see lines 30-32.
Line 29; “allowing for selectivity index 29 values (SIs) up to 11.58; instead, the BPPB concentration causing 50% hemolysis (HC50) was found 16-173 and 4-192-fold higher than the IC50 calculated for MeOV and MeTRAV PLX-R cells, respectively” The sentence is not undesrstandable. What for is the Sls index here? How it correlates to HC50 value, or maybe to IC50 value? How to interpret the number results: 16-173 and 4-192. This part must be rewritten.
To answer the Reviewer, the SIs values refer to CMM cells. This information has been added (line 35). Concerning the other comment by the Reviewer, we did not assert any correlation between SIs values for cancer cells and HC50 value, but we only declared that the concentration of BPPB necessary to kill 50% of red blood cells (HC50, Haemolytic concentration) was several times inferior to that necessary to kill 50% of CMM cells (IC50, Inhibition concentration). Anyway, for more clarity, these sentences were rewritten. Please, see lines 36-38.
Line 181, 302; There is lack in the caption figure meaning of the dots on the graphs.
Thank you so much for your suggestion. The missing specification has been included in all the Figure captions of graphs showing dots.
Line 449; There is no graph a) and c). What more, this section, regarding the cytotoxicity should have been displayed after the figure 1, where the cell viability of MeOV and MeTRAV PLX-R CMM was analyzed. It is mandatory to compare the cytotoxic efrect either to cancerogenic or normal cell line.
Why the Reviewer assert that in the original version of our manuscript graphs a) and b) are inexistent? Old Figure 6 (original version) was composed of three panels: a) a bar graph, b) a line graph and c) a dispersion graph with best fitting nonlinear models. In this revised manuscript, on suggestions by another Reviewer, Figure 6 has been modified, and panel (c) has been removed and moved to the Supplementary Materials file, newly created.
Concerning the position of the discussion the Reviewer refers to, her/his suggestion is not adequate to our organizational intentions. In our project, we thought clearer first dedicate an entire section to the sole discussion of the effects of PBBP on tumour cells, and then a separate section to the discussion of BPPB effects to HaCaT normal cells, only. The comparison between the effects of BPPB on cancerogenic and normal cell was reported after having analysed separately those towards cancer and normal cells in an additional separate section 2.2.7. On these considerations, we kindly ask the Reviewer to not force us to change the rational (in our opinion) organization of the manuscript.
Line 663; It is lack of the describing in details how the ROS production assay was performed. Explain it.
We thank the Reviewer for her/his suggestion. We have added the information asked. Please, see lines 739-745.
Reviewer 2 Report
Comments and Suggestions for Authors
The article by Alfei et al delives an interesting approach in CMM therapeutic approach by employing TPP-based BPPB as chemotherapeutic against MeOV and BeTRAV cells. Although the article itself is interesing and might provide new insights, in its current state should be rejected.
Major issues:
- The graphs presented throughout the manuscript seem to be performed using different software sets, making figures look bizarre. Please unify the graphing software used to generate the graphs, preferably use Prism instead of Excel.
- Are Figures 2a and 3a necessary? They are representing the same data as Figure 1. Consider merging subfigures 2b and 3b into one figure and drop subfigures 2a and 2b.
- Figures 6a and 7a are not present
- The overall quality of Figures is rather poor. Please upload higher-resolution graphs
- Although keratinocytes were used to examine the safety of employed compound on the potentially surrounding cells, it would be informative to also examine fibroblasts in this case
- Figure 4 states that H2O2 production by cells was examined, however the ratio of H2O2-positive to live cells was shown. Also, how were live cells determined in this assay? As H2O2 might be a secondary product of reactions were oxygen free radicals are present, the results shown here might be prone to overestimation. I'd recommend that raw fluorescence data be presented here.
- Please explain the rationale of the use of different fitting functions in the manuscript
- l. 342-344 It is rather a strong statement claiming that the correlation above 0.9 can be considered very poor
- What kind of medium - low-glucose or high-glucose was used for HaCaT culture? It might significantly influence the result of the therapy
Minor issues:
- Figure 5A seems to be cut in both x and y axis. Also, figure 5c should have its legend below and not above the graph
- l. 41 - replace "that originate" to "originating" to avoid repetition of "that" in the same sentence
- l. 50 and 51 - replace one of the "especiallly" with a synonym
- The citations are not consistently shown in the manuscript text - sometimes they are glued to the former word and sometimes there is a space left. Please unify. These kind of grammatical errors are quite common in the text regarding commas and full-stops.
- l. 133 "rational" should be replaced with "rationale"
- e.g. l. 259 and 249 are lacking the closing parentheses
- Were white, transparent or black-bottomed plates used for cell viability assays?
Author Response
The article by Alfei et al delives an interesting approach in CMM therapeutic approach by employing TPP-based BPPB as chemotherapeutic against MeOV and BeTRAV cells. Although the article itself is interesing and might provide new insights, in its current state should be rejected.
Major issues:
- The graphs presented throughout the manuscript seem to be performed using different software sets, making figures look bizarre. Please unify the graphing software used to generate the graphs, preferably use Prism instead of Excel.
We thank the Reviewer for her/his suggestion, which we have followed. Anyway, we want to explain that, although we recognize that two different software have been used, Excel and PRIS, as specified in the main text and/or in the Figure captions, it was necessary for the correlation study reported in our work. Precisely, we investigated the existence of possible correlations between the effects of BPPB and its concentrations, the treatments times and ROS production, based on the R2 of the regression models which best fitted the various set of data. In this regard, differently from PRISM, Excel allows to fit several types of regression models to selected data sets, such as linear, polynomial, exponential, logarithmic, and power, providing for all of them the related equations and R2 values. This function has permitted us to choose the best fitting models, among the more statistical relevant ones and to detect if any correlations existed based on the value of R2 which should be > 0.95.
Anyway, we agree with the Reviewer, that the existence of two types of graphs representations could negatively affect the aspect of the manuscript, making it “bizarre”. So, to avoid this inconvenience and follow the Reviewer suggestion of making uniform graphs, we have created a new Supporting Materials file, where all graphs created using Excel have been moved. In this way, both the manuscript and the Supporting Materials appear uniform now. The main text and Figure captions have been changed accordingly. We hope that Reviewer is satisfied with this solution.
- Are Figures 2a and 3a necessary? They are representing the same data as Figure 1. Consider merging subfigures 2b and 3b into one figure and drop subfigures 2a and 2b.
As asked subfigures 2b and 3b were merged in a single Figure 2, where they appear as panel (a) and (b). Figures 2a and 3a have been removed.
- Figures 6a and 7a are not present
We apologise in advance to the Reviewer, but after a careful checking of the original version of our manuscript submitted to IJMS, we are confident to confirm that Figure 6a and 7a were present. However, it is of little importance now because all Figures, their sequence and numbering has been transformed to address the first point of this revision.
- The overall quality of Figures is rather poor. Please upload higher-resolution graphs
High resolution graphs in line with the requirements of IJMS have been uploaded.
- Although keratinocytes were used to examine the safety of employed compound on the potentially surrounding cells, it would be informative to also examine fibroblasts in this case.
We thank the Reviewer for this comment, which has given us the possibility to better clarify our choices. Since about 90% of epidermis consists of keratinocytes and the remaining 10% is represented by melanocytes, and since we aim at a future development of BPPB as a topical treatment to cure the melanoma skin lesions, we have decided to use keratinocytes to examine the safety of the BPPB on healthy skin cells. Anyway, the request of the Reviewer can find satisfaction in our previous work (https://doi.org/10.3390/nano14181505). In fact, the cytotoxicity of BPPB towards other mammalian cells including human hepatocytes (HepG2), monkey kidney cells (Cos-7) and human fibroblast (MRC-5), specifically requested by the Reviewer, was already investigated in Alfei et al (https://doi.org/10.3390/nano14181505) published in Nanomaterials (2024). Anyway, to further satisfy the Reviewer, results from these previous investigations have been included in Table S1 and S2 in Supplementary Materials. Additional information on this question has been included in the test. Please, see lines 479-488.
- Figure 4 states that H2O2 production by cells was examined, however the ratio of H2O2-positive to live cells was shown. Also, how were live cells determined in this assay? As H2O2 might be a secondary product of reactions were oxygen free radicals are present, the results shown here might be prone to overestimation. I'd recommend that raw fluorescence data be presented here.
We apologize for this misunderstanding, but, during the preparation of these graphs, we have wrongly reported "DCFH positive cells (%)/Live cells (%)" on the ordinate axis instead of "DCFH positive cells (%)". In fact, as explained in Material and Methods section of the revised version, the values of fluorescence were normalized to the protein content. All graphs have been corrected.
Although we understand the doubt of the Reviewer, we believe that the results of ROS production herein reported were not overestimated since the method used is widely accepted. It utilizes DCFH-DA, a non-fluorescent compound which, after diffusion into the cells, is deacetylated by cellular esterases to 2’,7’-dichlorofluorescein which, in presence of ROS and in particular of H2O2 , the most long-lived ROS, is converted in the oxidized highly-fluorescent molecule DCF, detectable by fluorimetric analyses [Sies, H.; Belousov, V.V.; Chandel, N.S.; Davies, M.J.; Jones, D.P.; Mann, G.E.; Murphy, M.P.; Yamamoto, M.; Winterbourn, C. Defining roles of specific reactive oxygen species (ROS) in cell biology and physiology. Nat. Rev. Mol. Cell Biol. 2022, 23, 499–515; Smolyarova, D.D.; Podgorny, O.V.; Bilan, D.S.; Belousov, V.V. A guide to genetically encoded tools for the study of H2O2. FEBS J. 2022, 289, 5382–5395]. The suggestion of the Reviewer to report the raw fluorescence data in our opinion is not applicable since an increase of fluorescence could be due to a high number of cells but also to an increased production of H2O2 induced by the treatments. This is the reason why it is necessary to normalize the fluorescence data to the protein content.
- Please explain the rationale of the use of different fitting functions in the manuscript
We used different fitting functions in our study, since we aimed at finding the existence of possible correlations between the effects of BPPB and its concentrations, the treatments times and ROS production, as well as between ROS increase and BPPB concentration and time exposure, for both CMM cells and normal human HaCaT. This information was necessary to assume the existence of concentration-, time- or ROS-dependent mechanisms governing the cytotoxic effects of BPPB and its effects on ROS generation. To this end, all different fitting functions (or better regression models) provided by Excel software (linear, exponential, logarithmic, polynomial and power) were first applied to all couples of data sets considered, to detect the best fitting one on the base of R2 values of all models tested (provided by Excel software). Then, based on statistical indication of literature, it was possible to claim the actual existence of correlations, only when R2 was > 0.95.
- l. 342-344 It is rather a strong statement claiming that the correlation above 0.9 can be considered very poor.
As anticipated in the previous point and already reported in the original version of the main text ”R2 is an index that measures the goodness-of-fit of a statistical model applied to two series of data and its capability to correctly predit a new response variable “y” (cell viability %) from an new experimental data “x” (BPPB concentration) (https://doi.org/10.1002/bimj.19620040313). Higher the value of R2, higher the goodness-of-fit of the regression model used and higher the correlation existing between the two series of data fitted with that model. Low values of R2 also for the best fitting regression models indicated a low reliability of the models and a low correlation between data. In scientific studies, R2 of a model should be > 0.95 to consider that model reliable and to assert the existence of a certain correlation between data (https://www.statology.org/good-r-squared-value/).
- What kind of medium - low-glucose or high-glucose was used for HaCaT culture? It might significantly influence the result of the therapy
HaCat cell culture is carried out utilizing a medium with high-glucose (line 723).
Minor issues:
- Figure 5A seems to be cut in both x and y axis. Also, figure 5c should have its legend below and not above the graph
The Reviewer requests have been addressed. The original Figure 5a (4a in the revised version) has been corrected and legend of Figure 5c (4b in the revised version) has been moved below the graph.
- l. 41 - replace "that originate" to "originating" to avoid repetition of "that" in the same sentence
Done (line 48).
- l. 50 and 51 - replace one of the "especiallly" with a synonym
Done (line 57).
- The citations are not consistently shown in the manuscript text - sometimes they are glued to the former word and sometimes there is a space left. Please unify. These kind of grammatical errors are quite common in the text regarding commas and full-stops.
The Reviewer is right, and we are aware of these problems, but they do not depend on us, but on Mendeley software used to insert bibliography. We have already experimented this inconvenient, but the Editorial Office will adjust this tricky in the final version for publication.
- l. 133 "rational" should be replaced with "rationale"
Done (line 144).
- e.g. l. 259 and 249 are lacking the closing parentheses
The missing parentheses have been added.
- Were white, transparent or black-bottomed plates used for cell viability assays?
As specified in the text of the revised manuscript, transparent plastic plates have been used (line 695).
Round 2
Reviewer 2 Report
Comments and Suggestions for Authors
Dear Authors,
thank You for providing a revised version of the manuscript and addressing all of the raised issues. After the extensive review only one question remained - why ROS fluorescence was normalized to protein instead of cell number?
Author Response
We thank a lot the Reviewer for the further work of revision. Indeed, the question raised by the Reviewer is a thoughtful one and we apologize for being imprecise. Following the question of the Reviewer and our replay.
Dear Authors,
thank You for providing a revised version of the manuscript and addressing all of the raised issues. After the extensive review only one question remained - why ROS fluorescence was normalized to protein instead of cell number?
When ROS number in monitored by DCFH DA test, after the measurements, the standardization of ROS production as a function of cell number is necessary. Anyway, fluorescence values are usually normalized to the protein content, to avoid data misinterpretation due to variations in cell numbers (https://doi.org/10.1016/B978-0-12-416618-9.00013-3). Specifications on this question have been included in the main text with tha related reference (Ref. 44). Please, see lines 689-693.